# Four New Depsides Isolated from *Salvia miltiorrhiza* and Their Significant Nerve-Protective Activities

**DOI:** 10.3390/molecules23123274

**Published:** 2018-12-11

**Authors:** Qinghao Jin, Xinyi Hu, Yanping Deng, Jinjun Hou, Min Lei, Hongjian Ji, Jing Zhou, Hua Qu, Wanying Wu, Dean Guo

**Affiliations:** 1Shanghai Institute of Materia Medica, Chinese Academy of Sciences, Haike Road #501, Shanghai 201203, China; jinqinghao@simm.ac.cn (Q.J.); xy7215468@126.com (X.H.); shimbiro@163.com (Y.D.); jinjun_hou@simm.ac.cn (J.H.); mlei@simm.ac.cn (M.L.); hongjianji2006@163.com (H.J.); zj4603007@163.com (J.Z.); quhua@simm.ac.cn (H.Q.); 2Institute of Bioengineering, Zhejiang University of Technology, Hangzhou 310014, China

**Keywords:** *Salvia miltiorrhiza*, depsides, nerve-protection, HS-SY5Y

## Abstract

By investigating of the roots of *Salvia miltiorrhiza*, which is one of the most widely used Chinese herbs, we used phytochemical methods successfully to obtain twelve depsides: four depsides (**1**–**4**) that were previously undescribed, along with eight known ones (**5**–**12**). Their structure characteristics were assessed by HR-ESIMS, CD, NMR (^1^H, ^13^C, HSQC, HMBC) data analyses. These four newly isolated compounds (**1**–**4**), as well as the other eight compounds (**5**–**12**), show extraordinary protective effects on hydrogen peroxide-induced apoptosis in HS-SY5Y cells. Among them, depside **4** and depside **6** displayed more obviously protective effects than others.

## 1. Introduction

The hairy culture of *Salvia miltiorrhiza* is an important ingredient of traditional Chinese medicinal herb, which is named Danshen or Tanshen in Chinese. It has been widely used for the prevention as well as treatment of cardiovascular and cerebrovascular diseases [1] and osteoporosis [2]. Due to its high medicinal value, *S. miltiorrhiza* is drawing the attention of scientists nowadays. This herb has been officially recorded in the Chinese Pharmacopoeia since 1963. More than 2300 articles focused on *S. miltiorrhiza* can be found in the web of ScienceDirect published since 2009. In addition, many chemical compositions have been isolated from crude *S. miltiorrhiza*, typically divided into two categories. Phenolic acids are the most abundant of its hydrophilic compounds [3,4,5], and lipophilic chemicals such as Tanshinone [6,7], also play an important part in the research of this herb’s phytochemical and pharmacological aspects. The complex chemical compositions of *S. miltiorrhiza*, especially Salvianolic acid B and its ester derivatives, present significant bio-activities; their primary pharmacological activities have been found to be anti-oxidative [8], anti-inflammatory [9], anti-tumor [10] both in vivo and in vitro.

As is known to all, Alzheimer’s disease is a kind of degenerative brain disease, it is defined by the World Health Organization as “the disease or injury which initiated the train of events leading directly to death” [11]. To find an innovative way to treat Alzheimer’s disease more effectively, in the research of Lee, Y. W et al. [12], they tried to dig out a solution by turning to traditional Chinese herbs that had been ignored for such a long time. *S. miltiorrhiza* extract produced neuroprotection against Aβ_25–35_ peptide-induced apoptosis in SH-SY5Y cells. Additionally, in an article published in 2011 [13], total polyphenols and total tanshinones extracted from *S. miltiorrhiza* can work against toxicity mediated by Aβ_25–35_-induced cell viability decrease and apoptosis. Our group further isolated several compounds **1**–**12** (see Figure 1) from the 60% EtOH extraction of *S. miltiorrhiza*, which includes four new compounds **1**–**4**, together with several known compounds **5**–**12**. Their structure messages were determined by extensive spectroscopic analysis, including 1D and 2D NMR (^1^H, ^13^C, HSQC, HMBC), CD spectra, HR-ESI-MS. From the experiment of hydrogen peroxide-induced apoptosis SH-SY5Y human neuroblastoma cells, we found that the newly isolated compounds **1**–**4** both showed extraordinary protective effects; in particular, compound **4** had the most outstanding performance. Since oxidative neuro cells are the key factors in the development of neuro-degenerative diseases [14], we can speculate that the compounds isolated from *S. miltiorrhiza* may provide an effective way of the therapy of neuro-degenerative diseases.

## 2. Results and Discussion

Compound **1** was obtained as a yellowish amorphous powder. Its molecular formula was established as C_36_H_30_O_16_ by its HRESIMS of the [M − H]^−^ peak at *m/z* 717.1476. The ^1^H NMR spectral data of **1** showed signals for three ABX system benzene ring at δ_H_ 6.76 (1H, br d, 1.5, H-2ʹ), 6.67 (1H, d, 8.5, H-5′), and 6.61 (1H, d, 8.5, 1.5, H-6′) in B ring; 6.89 (1H, br d, 1.5, H-2″), 6.70 (1H, d, 8.0, H-5″), and 6.65 (1H, m, H-6″) in C ring; 6.57 (1H, br d, 1.5, H-2′′′), 6.60 (1H, d, 8.0, H-5′′′), and 6.42 (1H, dd, 8.0, 1.5, H-6′′′) in D ring; one ABXX system benzene ring at δ_H_ 6.79 (1H, d, 8.5, H-5), and 7.12 (1H, d, 8.5, H-6); one olefinic protons δ_H_ 7.56 (1H, d, 16.0, H-7), and 6.32 (1H, d, 16.0, H-8); three oxymethine protons δ_H_ 5.00 (1H, m, H-8′), 5.92 (1H, d, 11.0, H-7″), and 4.37 (1H, t, 6.0, H-8′′′), two methene protons at δ_H_ 3.07 (1H, m, H-7′a), and 2.93 (1H, m, H-7′b), 2.47 (1H, dd, 13.5, 6.0, H-7′′′a) and 2.38 (1H, dd, 13.0, 6.0, H-7′′′b). The ^13^C NMR spectrum of **1** showed 36 carbon signals, and characteristic signals include four carbonyl signals, twenty-four benzene ring signals, two olefinic carbon signals, one methoxy group. In the HMBC experiment, the correlations between H-8 and H-8′/C-9′, H-8″ and H-8′′′/C-9′′′, H-8″/C-1, C-2, C-3 confirmed that compound **1** and Salvianolic acid Y have the same plane structure (see Figure 2). The coupling constant ^3^*J*_7″,8″_ = 11.0 Hz in the benzofuran ring inferred it to be *cis*-oriented (7″*R*, 8″*S* or 7″*S*, 8″*R*). The 1D, 2D-NMR, HRESIMS for **1** is available in Appendix A. The CD spectrum of compound **1** (see Figure 3) showed a cotton effect at 226 nm (+8.53) and 260 nm (−0.95), determined to be the (7″*R*, 8″*S*) *cis* isomer of Salvianolic acid B [15,16]. In addition, the conformation of C-8′ and C-8′′′ were substantiated by C-8′ and C-8′′′ chemical shifts δ_C_ 77.8 (C-8′) and 78.2 (C-8′′′), indicating the C-8′ and C-8′′′ (*S*) conformations. Thus, the structure of **1** was established as 8′,8′′′-*epi*-Salvianolic acid Y. And the CD spectra for **1** is shown in Appendix A.

Both compounds **2** and **3** have the molecular formula C_37_H_32_O_16_, as shown from their positive HRESIMS at *m/z* 731.1638 [M − H]^−^ in **2**; *m/z* 731.1633 [M − H]^−^ in **3** and on the interpretation of ^13^C NMR data, respectively. Detailed analysis of NMR data (HSQC, HMBC) revealed that both compounds **2** and **3** had the same chemical structure of Salvianolic acid B, except for the increase of methoxy group protons in Salvianolic acid B. The HMBC spectrum presented correlation signals from 9′-OCH_3_ (δ_H_ 3.60, s) was connected with C-9′ (δ_C_ 169.9) in **2** and 9′′′-OCH_3_ (δ_H_ 3.55, s)/C-9′′′ (δ_C_ 169.2) in **3**. The above outcomes suggested that methoxy groups were joined to C-9′ of **2** and C-9′′′ of **3**, respectively. Thus, **2** was elucidated as 9′-methyl-Salvianolic acid B, and **3** was 9′′′-methyl-Salvianolic acid B. The 1D, 2D-NMR, HRESIMS for **2** is available in Appendix A, and the 1D, 2D-NMR, HRESIMS for **3** is available in Appendix A.

Compound **4** was obtained as a yellowish amorphous powder. Its molecular formula was established as C_37_H_32_O_16_ by its HRESIMS of the [M − H]^−^ peak at *m/z* 731.1636. The ^1^H NMR spectral data of **4** showed signals for three ABX system benzene rings, one ABXX system benzene ring, one olefinic proton, three oxygenated methine protons, two methylene protons, one methoxy group. The ^13^C NMR spectrum of **4** revealed 37 carbon signals, including one methoxy group. Analysis of the NMR spectral data of **4** with those of **2** showed these to be different in the characteristic C-8′′′ (δ_H_ 5.00, 1H, dd, 8.5, 4.5; δ_C_ 74.2) functionality, and this difference shows irrefutably that the C-8′′′ in **4** was in *S* configuration. Thus, the structure of **4** was established as 8′′′-epi-9′-methyl-Salvianolic acid B. The 1D, 2D-NMR, HRESIMS for **1** is available in Appendix A.

Oxidative stress-induced cell damage has been associated with neuro-degenerative diseases’ progressing for a long time [17,18,19,20]. Additionally, H_2_O_2_ is a mainly reactive oxygen species, considered to be the common inducer of apoptosis in many different cell types, especially the nerve cells [21]. In addition, HS-SY5Y cell line has been used to evaluate the potential protective effects of drugs in many experiments in recent years [22,23].

During this experiment, the protective effects of the extract of *S. miltiorrhiza* were assessed by 3-(4,5-dimethylthiazol-2-yl)-2,5-diphenyltetrazolium bromide MTT colorimetric assay. The cells without pre-treatment were treated as the control. As is shown in Figure 4, H_2_O_2_ induced significant decreases in cell survival. When the dosing concentration was fixed at 1 μM, when comparing experimental groups and control groups and H_2_O_2_-induced cell apoptosis groups, the conclusion could be drawn that among the compounds isolated from *S. miltiorrhiza*, compounds **4** and **6** showed the most significant protective effects on human neuroblastoma cells, while compounds **3** and **5** performed fewer protective functions than compounds **4** and **6**, and compounds **1**, **2**, **8** and **10** provided slight effects in terms of restoring cell survival, and the other five compounds did not show obviously protective effects of the HS-SY5Y cell line. Compared with the protective effect of the known compound salvianolic acid B, depside **4** has an antioxidant effect that is no weaker than salvianolic acid B, which restored HSHY-5Y cell viability to about 70%. Further research is needed to approach a significative structure–activity relationship.

## 3. Materials and Methods 

### 3.1. General

The organic solvents we used in our plant material extract and column chromatography were all of analytical grade, and the organic solvents we used in semi-preparative HPLC, as well as the HPLC and high-resolution MS analytical sample preparation, were all of HPLC-grade solvents. The mobile phase used formic acid (ROE Scientific Inc., Waltham, MA, USA) as the peak shape modifier. We chose column chromatography silica gel (200–300 mesh, Qingdao Marine Chemical Co., Ltd., Qingdao, China) as the medium for separating target compounds. The purification of the crude extract was performed on an Agilent 1100 HPLC system (Agilent Technologies, Waldbronn, Germany), using a ZIRBAX SB-C_18_ (9.4 × 250 nm, 5 μm; Agilent Technologies, Palo Alto, CA, USA). High-resolution MS and CID-MS^4^ were carried out on a Xevo G2S QTOF system. CD spectra were determined on a JASCO J-815 spectropolarimeter. The 1D and 2D NMR data (^1^H, ^13^C, HSQC, HMBC) were recorded on a Bruker Avance III HD Ascend 500 MHz spectrometer (Bruker BioSpin AG, Fällanden, Switzerland).

### 3.2. Plant Material

The samples of *Salvia miltiorrhiza* were collected from Shandong province, P. R. China, in October 2015. The plant material was identified by Prof. De-an Guo, National Engineering Laboratory for TCM standardization Technology, Shanghai Institution of Materia Medica, Chinese Academy of Sciences, and the voucher specimen was deposited there.

### 3.3. Extraction and Isolation

The dry roots of *Salvia miltiorrhiza* (5.0 kg) were minced to powder and refluxed exhaustively with 60% EtOH (3 × 80 L) under reflux for 3 h, and then the concentrated extract of *S. miltiorrhiza* (497 g) was partitioned between EtOAc, *n*-BuOH, MeOH (5 × 10 L) to get the MeOH-soluble fraction. The MeOH-soluble fraction (300 g) was suspended in MeOH and subjected to chromatography (silica gel, 0.045–0.075 mm), eluted with the mixtures of CH_2_Cl_2_-MeOH, EtOAc, EtOAc-MeOH with increasing polarity to afford 14 fractions (Fr1–Fr14). To further isolate the mixture, we used TLC to guide the following steps. Then, the fraction Fr12 extraction (803 mg) was separated by Sephadex LH-20 with a stepwise gradient of CH_2_Cl_2_-MeOH gradient (10:1–1:1) to give the sub-fraction F1-F10. After being subjected to semi-preparative HPLC (MeOH-H_2_O (75:25), 3 mL/min), subfraction F4 (120 mg) was divided into 4 fractions, after being purified, **1** (2.3 mg, t_R_ 7.2 min), **2** (3.0 mg, t_R_ 8.8 min). Fraction Fr8 (38 g) provided **3** (2.7 mg, t_R_ 9.7 min), salvianolic acid E **5** (19 mg, t_R_ 12.4 min) by semi-preparative HPLC (MeOH-H_2_O with 0.1% HCOOH (55:45), 3 mL/min). Fraction Fr8 (1.03 g) was separated by Sephadex LH-20 (MeOH, 50%) to yield isosalvianolic acid C **8** (5.3 mg), **4** (2.5 mg). Fraction Fr7 was subjected to semi-preparative HPLC (MeOH-H_2_O (80:20 to 20:80), 3 mL/min) to obtain salvianolic acid B **6** (10 mg, t_R_ 7.9 min), salvianolic acid C **7** (23 mg, t_R_ 10.5 min); the semi-preparative HPLC (MeOH-H_2_O with 0.1% HCOOH (50:50), 3 mL/min) was used to obtain lithospermic acid **9** (19 mg, t_R_ 7.7 min). After being subjected to a macro pore resin AB-10 (40%MeOH), fraction Fr5 (44 g) gave sub-fraction F1–F8, the semi-preparative HPLC (ACN-H_2_O with 0.1% HCOOH (75:25 to 30:70), 3 mL/min) was used to purified sub-fraction F2 to obtain salvianolic acid D **10** (13 mg, t_R_ 8.6 min). Sub-fraction 7 (190 mg) was subjected to semi-preparative HPLC (ACN-H_2_O with 0.1% HCOOH (50:50), 3 mL/min), rosmarinic acid **11** (9 mg, t_R_ 11.0 min), salvianic aid A **12** (11.2 mg, t_R_ 1.7 min) were collected separately. The purity of the compounds was measured by HPLC as more than 90%. 

### 3.4. Compound Characterization

8′,8′′′-*epi*-Salvianolic acid Y (**1**): Yellowish amorphous powder; HRESIMS *m/z* 717.1476 [M − H]^−^ (calcd. for C_36_H_29_O_16_:717.1456); CD (MeOH): 226 nm (+8.53), 260 nm (−0.95); ^1^H-NMR (CD_3_OD, 500 MHz) and ^13^C-NMR (CD_3_OD, 125 MHz) data, see Table 1.

9′-methyl-Salvianolic acid B (**2**): Yellowish amorphous powder; HRESIMS *m/z* 731.1638 [M − H]^−^ (calcd. for C_37_H_32_O_16_:731.1612); ^1^H-NMR (DMSO-*d*_6_, 500 MHz) and ^13^C-NMR (DMSO-*d*_6_, 125 MHz) data, see Table 1.

9′′′-methyl-Salvianolic acid B (**3**): Yellowish amorphous powder; HRESIMS *m/z* 731.1633 [M − H]^−^ (calcd. for C_37_H_32_O_16_:731.1612); ^1^H-NMR (DMSO-*d*_6_, 500 MHz) and ^13^C-NMR (DMSO-*d*_6_, 125 MHz) data, see Table 1.

8′′′-*epi*-9′-methyl-Salvianolic acid B (**4**): Yellowish amorphous powder; HRESIMS *m/z* 731.1636 [M − H]^−^ (calcd. for C_37_H_32_O_16_:731.1612); ^1^H-NMR (DMSO-*d*_6_, 500 MHz) and ^13^C-NMR (DMSO-*d*_6_, 125 MHz) data, see Table 1.

### 3.5. HS-SY5Y Cell Culture and Cell Viability Assay

The protective effects of the isolated compounds were evaluated on the defense effect of SH-SY5Y human neuroblastoma cells (Chinese Academy of Science Committee Type Culture Collection Cell, Bank, Shanghai, China) against H_2_O_2_-induced cytotoxicity. The SH-SY5Y cells were maintained in MEM medium (Sigma-Aldrich, St. Louis, MO, USA) in a humid atmosphere of 5% CO_2_ and 95% air at 37 °C, then the cells were treated with 350 μM H_2_O_2_ for 24 h. The conventional MTT colorimetric assay was applied to measure the neuroblastoma cell viability, and we used the cells without pre-treatment as the control. The SH-HY5Y cells were plated into 96-well plates at a density of 1 × 10^4^ per well and incubated for 24 h. The experimental groups were maintained in MEM supplemented with 1 μM of the obtained depsides **1**–**12** at the same condition of the control groups for 15 min, then they were exposed to 350 μM H_2_O_2_ for 24 h. Each group was further incubated with MTT (Sigma-Alarich, St. Louis, MO, USA) for another 4 h, the supernatant was removed after that, the crystals were dissolved in DMSO. The optical density of each well was evaluated by a microplate reader (BIO-RAD Model 680, Laboratories, Hercules, CA, USA) at 492 nm.

## 4. Conclusions

In this study, four new depsides (**1**–**4**), as well as another eight known compounds (**5**–**12**), were isolated from *S. miltiorrhiza*. Their purity was measured by HPLC to be more than 90%. All assessments of the structure of compounds were based on extensive spectroscopic studies, along with MS and NMR analyses. Additionally, the protective effects of these compounds were evaluated by reduced hydrogen peroxide-induced apoptosis in HS-SY5Y cell abilities, the newly obtained compounds showed significant protective effects on this human neuroblastoma cell line. The results of this study can not only enrich the types of compounds obtained from *S. miltiorrhiza*, but can also help with expanding the range of search for drugs for the treatment of neurodegenerative diseases.

## Figures and Tables

**Figure 1 molecules-23-03274-f001:**
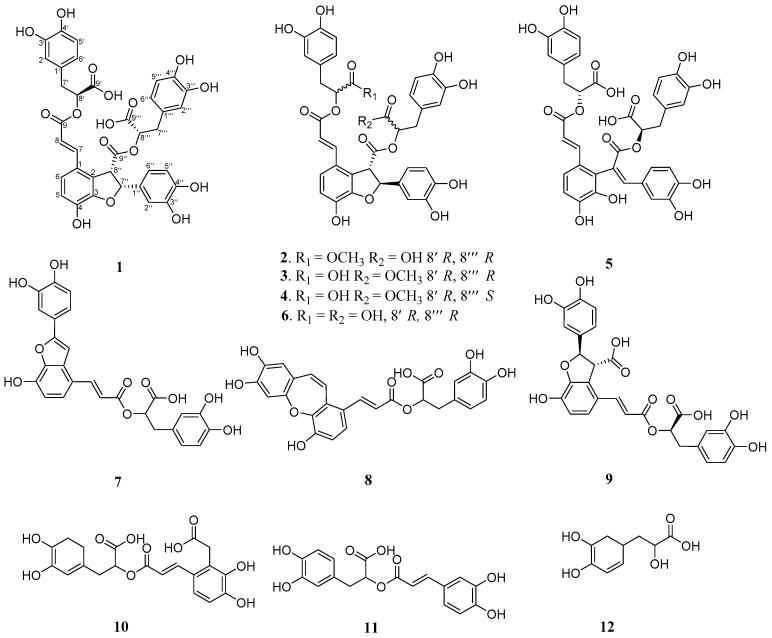
Chemical structures of compounds **1**–**12** from roots of *S. miltiorrhiza*.

**Figure 2 molecules-23-03274-f002:**
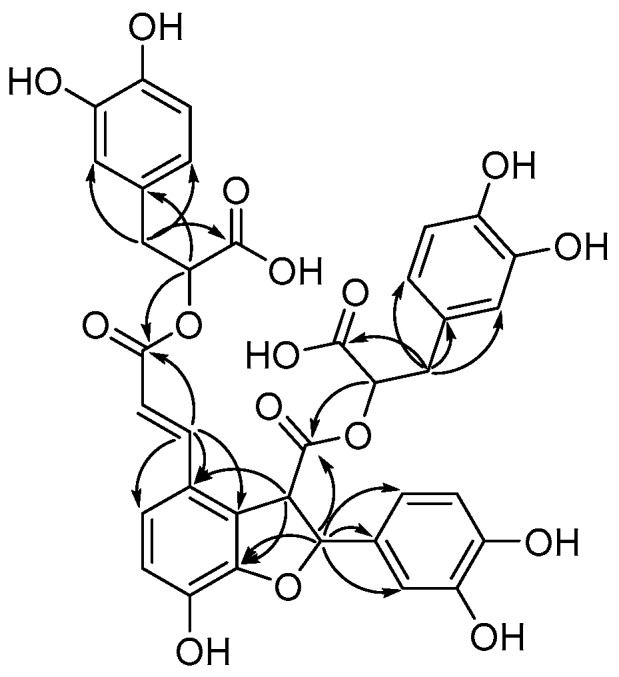
Key HMBC correlations (H→C) of compound **1**.

**Figure 3 molecules-23-03274-f003:**
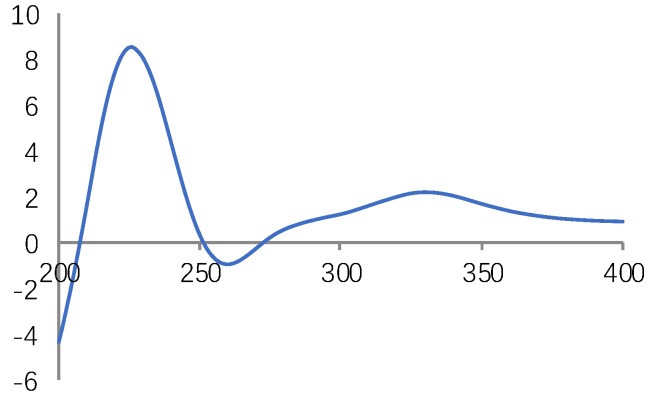
CD spectrum of compound **1**.

**Figure 4 molecules-23-03274-f004:**
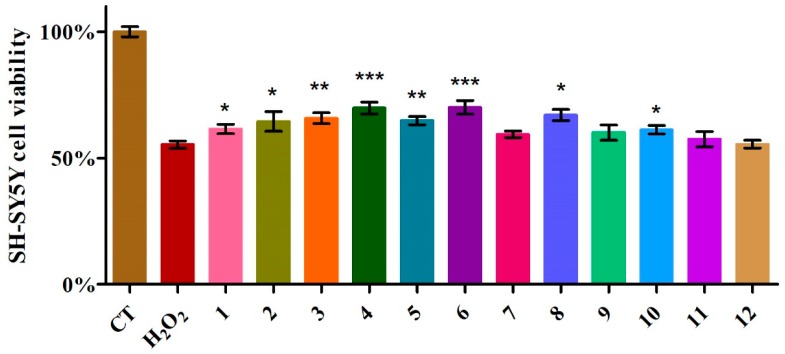
The protective effects of the different compounds (* means the *p* value compared with H_2_O_2_ group is lower than 0.05; ** means the *p* value compared with H_2_O_2_ group is lower than 0.01; *** means the *p* value compared with H_2_O_2_ group is lower than 0.001).

**Table 1 molecules-23-03274-t001:** ^1^H-(500 MHz) and ^13^C-NMR (125 MHz) data for the compounds **1**–**4** (^a^ is recorded in CD_3_OD; ^b^ is recorded in DMSO-*d*_6_).

C	1 ^a^		2 ^b^		3 ^b^		4 ^b^	
	δ_H_	δ_C_	δ_H_	δ_C_	δ_H_	δ_C_	δ_H_	δ_C_
1	-	124.9	-	122.7	-	115.6	-	122.5
2	-	128.0	-	124.8	-	124.9	-	124.9
3	-	149.6	-	147.3	-	147.2	-	147.1
4	-	144.8	-	143.8	-	144.2	-	134.9
5	6.79 d (8.5)	117.8	6.81 d (8.4)	116.7	6.83 d (8.4)	117.5	6.82 d (8.5)	117.3
6	7.12 d (8.5)	122.6	7.24 d (8.4)	121.7	7.29 d (8.4)	120.8	7.28 d (8.5)	121.1
7	7.56 d (16.0)	143.7	7.53 d (16.0)	142.9	7.52 d (15.8)	141.8	7.57 d (16.0)	142.7
8	6.32 d (16.0)	117.6	6.28 d (16.0)	114.9	6.34 d (15.8)	115.4	6.33 d (16.0)	114.8
9	-	169.3	-	165.9	-	165.8	-	165.7
1′	-	130.8	-	126.7	-	127.7	-	126.4
2′	6.76 br d (1.5)	117.6	6.66 br d (2.0)	117.2	6.66 brd (2.0)	116.4	6.65 br s	116.6
3′	-	144.8	-	145.6	-	144.9	-	144.9
4′	-	150.0	-	144.3	-	143.9	-	144.1
5′	6.67 d (8.5)	116.1	6.60 d (8.0)	115.7	6.62 d (8.0)	115.4	6.61 d (8.0)	115.4
6′	6.61 dd (8.5, 1.5)	121.8	6.36 br d (8.0)	119.5	6.52 dd (8.0, 2.0)	119.9	6.47, dd (8.0, 2.0)	120.0
7′	3.07 m; 2.93 m	38.2	2.90, 6.5 d	36.3	2.98 dd (14.0, 4.0)	36.3	2.92 br d (6.5)	36.2
		-	-	-	2.84 m	-	-	-
8′	5.00 m	77.8	-	73.1	-	73.5	-	73.0
9′	-	177.8	-	169.9	-	171.2	-	169.8
1″	-	129.3	-	131.5	-	131.2	-	131.2
2″	6.89 br d (1.5)	115.0	6.68 br d (2.0)	112.4	6.68 br d (2.0)	112.6	6.66 br d (1.5)	112.5
3′’	-	145.6	-	145.4	-	145.1	-	144.8
4′’	-	146.9	-	145.4	-	145.5	-	145.5
5′’	6.70 d (8.0)	116.1	6.72 d 8.0	115.6	6.72 d 8.0	115.6	6.72 d (8.0)	115.4
6′’	6.65 m	119.9	6.43 br d (8.0)	116.7	6.55 m	117.0	6.55 m	116.9
7′’	5.92 d (11.0)	88.4	5.67 br d (3.0)	85.8	5.65 br d (4.0)	86.0	5.65 brd (4.0)	85.9
8′’	4.76 d (11.0)	54.7	4.36 br d (3.0)	55.4	4.43 br d (4.0)	54.9	4.43 br d (4.0)	55.1
9′’	-	171.5	-	170.1	-	170.3	-	170.1
1′′′	-	129.5	-	128.1	-	126.2	-	127.0
2′′′	6.57 br d (1.5)	118.1	6.55 m	116.4	6.55 m	116.5	6.58 br d (2.0)	116.4
3′′′	-	144.9	-	144.9	-	143.9	-	145.4
4′′′	-	145.0	-	144.1	-	144.2	-	145.1
5′′′	6.60 d (8.0)	121.7	6.57 d (8.0)	115.6	6.57 d (8.0)	115.6	6.54 d (8.0)	115.5
6′′′	6.42 dd (8.0, 1.5)	122.4	6.34 br d (8.0)	120.0	6.32 m	120.1	6.31 dd (8.0, 2.0)	120.0
7′’’	2.38 dd (13.5, 6.0) 2.47 dd (13.5, 6.0)	37.7	2.95 dd (14.0, 3.0) 2.80 dd (14.0, 9.0)	36.3		35.8	2.86 dd (14.0, 4.5) 2.82 dd (14.0, 8.0)	35.9
8′′′	4.37 t (6.0)	78.2	4.93 dd (9.0, 3.0)	75.3	5.14 dd (8.0, 5.0)	73.9	5.00 dd (8.5, 4.5)	74.2
9′′′	-	177.5	-	170.7	-	169.2	-	170.1
9′-OCH_3_	-	-	3.60 s	52.0	-	-	3.58 s	51.0
9‴-OCH_3_	-	-	-	-	3.55 s	51.9	-	-

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
