# Peer review of "Four New Depsides Isolated from Salvia miltiorrhiza and Their Significant Nerve-Protective Activities"

_molecules, 2018, doi:10.3390/molecules23123274_

Reviewer 1 Report

This paper reported the isolation and purification of four new depsides (1-4) and 8 known depsides (5-12) from the 60% EtOH extract of Salvia miltiorrhiza roots. These deposides (1 µM) were examined for the neuroprotection activity against H2O2-induced neuronal damage of SH-SY5Y human neuroblastoma cells, showing more obviously protective effects of depside 4 and 6 than others. There are some concerns as listed in the following:

*L48: isolated several compounds 1-12 from the water extraction of S. miltiorrhiza vs. L146: with 60% EtOH (3 × 80 L) under reflux

L54: oxidative? is a key factor

L109: of the extract? of the of S. miltiorrhiza

*L120: Figure 4, no known positive compound for comparison?

L150: 14 fractions(Fr1-Fr14)

L151: Fr12extraction(803 mg)

L160: 9(19 mg, tR 7.7 min)After subjected to

*L180-182: [The protective effects of the isolated compounds were evaluated on the anti-apoptosis in SH-SY5Y human neuroblastoma cells pre-treated by hydrogen peroxide (Chinese Academy of Science Committee Type Culture Collection Cell, Bank, Shanghai, China).]

(1) anti-apoptosis? (only MTT assay cannot make sure whether the protect effect of the test compound is due to anti-apoptosis)

(2) pre-treated by hydrogen peroxide (it is suggested that the test compound is post-treated?) -> Please clearly describe the test compound is treated before or at same time, or after hydrogen peroxide addition.

*L166: Please add the name of despides 5-12.

L188: the different extractions? of

L215: Please check all inconsistent format for journal abbreviation, e.g. R4: J. N. Pro., R7: J. N. Prod., R23: J. Ethnopharmacology

*It is unclear whether the protective potency of the present depside 4 or depside 6 is better than any previous reported compounds (such as Salvianolic acidis etc.) isolated from Salvia miltiorrhiza for against H2O2-induced neuronal damage. A recent paper has reviewed In vitro and in vivo neuroprotective effects of Salvianolic acids (Int. J. Mol. Sci. 2018, 19, 458; doi:10.3390/ijms19020458).

Author Response

Reviewers' comments: Reviewer #1: A few concerns: 1) *L48:isolated several compounds 1-12 from the water extraction of S. miltiorrhiza vs. L146: with 60% EtOH (3 × 80 L) under reflux. Answer: Authors’s thank for reviewer’s comment, authors want to express their sincerely apology to these rookie mistake. authors revised the manuscript L48 as following: “Our group further isolated several compounds 1-12 from the 60% EtOH extraction of S. miltiorrhiza” in P2 L51. It can be confirmed that this experiment was based on the 60%EtOH extraction of S. miltiorrhiza. 2) oxidative? is a key factor Answer: As what the comment pointed out, and to make it clear, authors revised the article as “Since oxidative neuro cells are the key factors in the development of neuro-degenerative diseases” in P2 L57. 3) of the extract? of the of S. miltiorrhiza Answer: As what the comment pointed out, authors revised the article as “the protective effects of the extract of S. miltiorrhiza” in P5 L110. 4) Figure 4, no known positive compound for comparison? Answer: To compared the protective effects of the depsides, authors turned to compared them to the known positive compounds salvianolic acid B, aiming at proving the newly depsides’ significant neuro-protective activity. The salvianolic acid B’s group was treated as the experiment groups, after the incubation of H2O2, the HSHY-5Y cell viability restored to about 70%, equivalented to the experimental data of depside 4. As what the comment pointed out, authors revised the article as “Compared with the protective effect of the known compound salvianolic acid B, depside 4 has an antioxidant effect that is not weaker than salvianolic acid B, which made the HSHY-5Y cell viability restore to about 70%” in P5 L119. 5) 14 fractions(Fr1-Fr14) Answer: As what the comment pointed out, authors revised the article as “14 fractions (Fr1-Fr14)” in P6 L153. 6) L151: Fr12extraction(803 mg) Answer: As what the comment pointed out, authors revised the article as “Then the fraction Fr12 extraction (803 mg)” in P6 L154. 7) L160: 9(19 mg, tR 7.7 min)After subjected to Answer: As what the comment pointed out, authors revised the article as “obtain lithospermic acid 9 (19 mg, tR 7.7 min).After subjected to” in P7 L163. 8) *L180-182: [The protective effects of the isolated compounds were evaluated on the anti-apoptosis in SH-SY5Y human neuroblastoma cells pre-treated by hydrogen peroxide (Chinese Academy of Science Committee Type Culture Collection Cell, Bank, Shanghai, China).] (1) anti-apoptosis? (only MTT assay cannot make sure whether the protect effect of the test compound is due to anti-apoptosis) Answer: As what the comment pointed out, It was said that “anti-apoptosis” is inaccurate and should be “neuroprotective”. So authors revised it as “The protective effects of the isolated compounds were evaluated on the defense effect of SH-SY5Y human neuroblastoma cells (Chinese Academy of Science Committee Type Culture Collection Cell, Bank, Shanghai, China) against H2O2-induced cytotoxicity” in P7 L185. (2) pre-treated by hydrogen peroxide (it is suggested that the test compound is post-treated?) -> Please clearly describe the test compound is treated before or at same time, or after hydrogen peroxide addition. Answer: As what the comment pointed out, to make it clear, authors revised the article as “The experimental groups were maintained in MEM supplemented with 1 μM of the obtained depsides 1-12 at the same condition of the control groups for 15 min, then them were exposed to 350 μM H2O2 for 24 h” in P7 L192. 9) *L166: Please add the name of despides 5-12. Answer: To make it clearly, authors added the names of despides 5-12 in P7 L158-L168. 10) L188: the different extractions? Of Answer: To make it clearly, authors revised it as “of the obtained depsides 1-12” of P7 L193. 11) L215: Please check all inconsistent format for journal abbreviation, e.g. R4: J. N. Pro., R7: J. N. Prod., R23: J. Ethnopharmacology Answer: According to the comments, authors revised the R4 journal abbreviation as “J. N. Prod.” and revised R23 journal abbreviation as “J.Ethnopharmacol”.

Reviewer 2 Report

The authors submitted very interesting manuscript dealing with caffeic acid derivatives which were isolated form extracts of Salvia miltiorrhiza. Furthermore, the authors investigated also neuroprotectve effects of these derivatives.

The authors use adequate methods. I appreciate especially the phytochemical part of the manuscript because isolation and purification of plant constituents require often a lot of time and patience. However, the used methods for neuroprotective effects should be described better in the part of Materials and Methods.

I recommend the manuscript for the publication after some revision as mentioned above.

Author Response

 1)       *L48:isolated several compounds 1-12 from the water extraction of S. miltiorrhiza vs. L146: with 60% EtOH (3 × 80 L) under reflux.

Answer: Authors thank for reviewer’s comment, authors want to express their sincerely apology to these rookie mistake. authors revised the manuscript L48 as following: “Our group further isolated several compounds 1-12 from the 60% EtOH extraction of S. miltiorrhiza in P2 L51. It can be confirmed that this experiment was based on the 60%EtOH extraction of S. miltiorrhiza.

2)    oxidative? is a key factor

Answer: As what the comment pointed out, and to make it clear, authors revised the article as “Since oxidative neuro cells are the key factors in the development of neuro-degenerative diseases” in P2 L57.

3)    of the extract? of the of S. miltiorrhiza

Answer: As what the comment pointed out, authors revised the article as “the protective effects of the extract of S. miltiorrhiza” in P5 L110.

4)    Figure 4, no known positive compound for comparison?

Answer: To compared the protective effects of the depsides, authors turned to compared them to the known positive compounds salvianolic acid B, aiming at proving the newly depsides’ significant neuro-protective activity. The salvianolic acid B’s group was treated as the experiment groups, after the incubation of H2O2, the HSHY-5Y cell viability restored to about 70%, equivalented to the experimental data of depside 4. As what the comment pointed out, authors revised the article as “Compared with the protective effect of the known compound salvianolic acid B, depside 4 has an antioxidant effect that is not weaker than salvianolic acid B, which made the HSHY-5Y cell viability restore to about 70%” in P5 L119.

5)    14 fractions(Fr1-Fr14)

Answer: As what the comment pointed out, authors revised the article as “14 fractions (Fr1-Fr14)” in P6 L153.

6)    L151: Fr12extraction(803 mg)

Answer: As what the comment pointed out, authors revised the article as “Then the fraction Fr12 extraction (803 mg)” in P6 L154.

7)    L160: 9(19 mg, tR 7.7 min)After subjected to

Answer: As what the comment pointed out, authors revised the article as “obtain lithospermic acid 9 (19 mg, tR 7.7 min).After subjected to” in P7 L163.

8)    *L180-182: [The protective effects of the isolated compounds were evaluated on the anti-apoptosis in SH-SY5Y human neuroblastoma cells pre-treated by hydrogen peroxide (Chinese Academy of Science Committee Type Culture Collection Cell, Bank, Shanghai, China).]

(1) anti-apoptosis? (only MTT assay cannot make sure whether the protect effect of the test compound is due to anti-apoptosis)

Answer: As what the comment pointed out, It was said that “anti-apoptosis” is inaccurate and should be “neuroprotective”. So authors revised it as “The protective effects of the isolated compounds were evaluated on the defense effect of SH-SY5Y human neuroblastoma cells (Chinese Academy of Science Committee Type Culture Collection Cell, Bank, Shanghai, China) against H2O2-induced cytotoxicity” in P7 L185.

(2) pre-treated by hydrogen peroxide (it is suggested that the test compound is post-treated?) -> Please clearly describe the test compound is treated before or at same time, or after hydrogen peroxide addition.

Answer: As what the comment pointed out, to make it clear, authors revised the article as “The experimental groups were maintained in MEM supplemented with 1 μM of the obtained depsides 1-12 at the same condition of the control groups for 15 min, then them were exposed to 350 μM H2O2 for 24 h” in P7 L192.

9)    *L166: Please add the name of despides 5-12.

Answer: To make it clearly, authors added the names of despides 5-12 in P7 L158-L168.

10) L188: the different extractions? Of

Answer: To make it clearly, authors revised it as “of the obtained depsides 1-12” of P7 L193.

11) L215: Please check all inconsistent format for journal abbreviation, e.g. R4: J. N. Pro., R7: J. N. Prod., R23: J. Ethnopharmacology

Answer: According to the comments, authors revised the R4 journal abbreviation as “J. N. Prod.” and revised R23 journal abbreviation as “J.Ethnopharmacol”.

Round  2

Reviewer 2 Report

Dear Authors and Editor,

I read the revised version of manuscript and all supplementary materials. The quality of this revised manuscript substantially improved. Therefore, I am pleased that I may recommend the manuscript for publication.